# Deep learning in rare disease. Detection of tubers in tuberous sclerosis complex

Iván Sánchez Fernández[1]*, Edward Yang[1], Paola Calvachi[2], Marta Amengual-Gual[1], Joyce Y. Wu[3], Darcy Krueger[4], Hope Northrup[5], Martina E. Bebin[6], Mustafa Sahin[1], Kun-Hsing Yu[7], Jurriaan M. Peters[1], on behalf of the TACERN Study Group¶

1 Boston Children's Hospital, Harvard Medical School, Boston, MA, United States of America, 2 Brigham and Women's Hospital, Harvard Medical School, Boston, MA, United States of America, 3 Mattel Children's Hospital, David Geffen School of Medicine at University of California Los Angeles, Los Angeles, CA, United States of America, 4 Cincinnati Children's Hospital Medical Center, Cincinnati, OH, United States of America, 5 The University of Texas Health Science Center at Houston, Houston, Texas, United States of America, 6 University of Alabama at Birmingham, Birmingham, Alabama, United States of America, 7 Department of Biomedical Informatics, Harvard Medical School, Boston, MA, United States of America

¶ Membership of the TACERN Study Group is listed in the Acknowledgments.
* ivan.fernandez@childrens.harvard.edu

**Data Availability Statement:** All code and results are available. The original neuroimages are not directly available because public distribution of patient data has to be requested to the Institutional

## Abstract

### Objective

To develop and test a deep learning algorithm to automatically detect cortical tubers in magnetic resonance imaging (MRI), to explore the utility of deep learning in rare disorders with limited data, and to generate an open-access deep learning standalone application.

### Methods

T2 and FLAIR axial images with and without tubers were extracted from MRIs of patients with tuberous sclerosis complex (TSC) and controls, respectively. We trained three different convolutional neural network (CNN) architectures on a training dataset and selected the one with the lowest binary cross-entropy loss in the validation dataset, which was evaluated on the testing dataset. We visualized image regions most relevant for classification with gradient-weighted class activation maps (Grad-CAM) and saliency maps.

### Results

114 patients with TSC and 114 controls were divided into a training set, a validation set, and a testing set. The InceptionV3 CNN architecture performed best in the validation set and was evaluated in the testing set with the following results: sensitivity: 0.95, specificity: 0.95, positive predictive value: 0.94, negative predictive value: 0.95, F1-score: 0.95, accuracy: 0.95, and area under the curve: 0.99. Grad-CAM and saliency maps showed that tubers resided in regions most relevant for image classification within each image. A stand-alone trained deep learning App was able to classify images using local computers with various operating systems.

Review Board. There are legal and ethical restrictions on how to share clinical data. Medical researchers with proper training on the ethical management of de-identified clinical data should contact the Institutional Review Board at Boston Children's Hospital (IRB@childrens.harvard.edu) to request data access. All code with results and full models can be found at: https://ivansanchezfernandez.github.io/TSC_code_results_models/. The code on training and validation includes network convergence plot for each model showing the decrease in loss and increase in accuracy with plateauing, showing that the number of training examples was enough for training. In addition, we have released our best CNN model as an easy to use App with Kivy, version 1.10.1 (36), so that readers can test the model on their own MRI images. The App and instructions for Windows can be found at:Detection of tubers with convolutional neural networks, 12, https://ivansanchezfernandez.github.io/TSC_TuberFinder_Windows/. 265 The App and instructions for Apple can be found at: https://ivansanchezfernandez.github.io/TSC_TuberFinder_Apple/.

**Funding:** JMP, MS, HN, JYW, DK and MEB were supported by the National Institute of Neurological Disorders And Stroke of the National Institutes of Health (NINDS) and Eunice Kennedy Shriver National Institute of Child Health & Human Development (NICHD) under Award Number U01NS082320. ISF has received an Amazon Web Services Cloud Credits for Research support in the form of computational credits for his project on "Identification and localization of tubers in Tuberous Sclerosis Complex with deep learning convolutional neural networks". JYW, DK, HN, MEB, MS, and JP received funding to collect the data as a part of the TACERN collaborative. The funders had no additional role in study design, data analysis, decision to publish, or preparation of the manuscript. The specific roles of these authors are articulated in the 'author contributions' section.

**Competing interests:** ISF has received an Amazon Web Services Cloud Credits for Research support in the form of computational credits for his project on "Identification and localization of tubers in Tuberous Sclerosis Complex with deep learning convolutional neural networks". JYW, DK, HN, MEB, MS, and JP received funding to collect the data as a part of the TACERN collaborative. There are no patents, products in development or marketed products to declare. This does not alter our adherence to PLOS ONE policies on sharing data and materials.

## Conclusion

This study shows that deep learning algorithms are able to detect tubers in selected MRI images, and deep learning can be prudently applied clinically to manually selected data in a rare neurological disorder.

## Introduction

Tuberous sclerosis complex (TSC) is a genetic neurocutaneous syndrome with an incidence of 1/6,000 to 1/10,000 live births and a population prevalence of 1/12,000 to 1/25,000 [1–3]. The number, size, and morphology of tubers vary widely between individuals [4, 5] and, rarely, TSC can be clinically paucisymptomatic or with only subtle findings on MRI, such as a single lesion [6]. Automating tuber detection in brain MRIs can enhance diagnostic certainty in resource-rich areas and facilitate diagnosis in areas where medical specialists are not readily available.

Convolutional neural networks (CNNs) automatically detect patterns of interest in images and have demonstrated image-classification performance at or above the level of humans [7], including detection of diabetic retinopathy [8], skin cancer [9], echocardiography findings [10], and acute neuroimaging findings [11] at the level of specialist physicians. These studies required many thousands of images to train the CNNs, which are challenging to obtain in rare neurological disorders like TSC and make computerized support of rare disorders difficult to develop. We could not find studies using deep learning to detect tubers in TSC. The application of CNNs in clinical practice is also frequently limited because of privacy concerns.

This study aims to demonstrate that CNNs can be successfully developed for detection of rare brain anomalies on MRI such as tubers using a relatively small number of training images. It also aims to demonstrate how CNNs can be implemented using a thin-client model such that a clinician can use advanced classifier tools at the point of care without the need to transfer patient sensitive data to a third party computer system.

## Patients and methods

### Ethical approval

The Internal Review Board at Boston Children's Hospital approved this study (IRB-P00029015) and determined that it met the regulatory requirements to obtain a waiver of inform consent/authorization from research subjects as this study was a secondary use of already existing data collected primarily for clinical reasons.

### Study design

This study applies CNNs, a type of neuronal network developed for computer vision, to detect tubers in brain MRI images. For a more in-depth overview of CNNs, transfer learning, data augmentation, and visualization techniques relevant to this article, please see Supplementary Methods at: https://ivansanchezfernandez.github.io/TSC_supplementary_methods/. All supplementary files at dx.doi.org/10.17504/protocols.io.bdt3i6qn

## Patients

Our population of interest consists of children and adolescents with TSC and tubers visible on their MRI. Our representative sample consisted of patients with TSC followed at the Multidisciplinary Tuberous Sclerosis Clinic and controls with normal MRI from Boston Children's Hospital, a tertiary pediatric center with a heterogeneous sample of patients with TSC of a wide variety of ages, severities, and comorbidities. The inclusion criteria for patients with TSC were: 1) patients with a confirmed diagnosis of TSC following the 2012 International Tuberous Sclerosis Complex Consensus Conference Diagnostic Criteria [12], 2) who had at least one brain MRI at Boston Children's Hospital, 3) who had both T2-axial and FLAIR-axial sequences available, and 4) who had tubers detected clearly on MRI as per the radiology report during routine clinical care. The inclusion criteria for the control group were: 1) patients who had at least one brain MRI at Boston Children's Hospital, 2) the MRI was interpreted by the neuroradiologist as normal or with non-specific findings during routine clinical care. Common indications for obtaining an MRI in controls were headache, concussion, non-syndromic mild developmental delay, and treatment-responsive epilepsy. When several brain MRIs were available per patient, we selected the most recent MRI to minimize the decreased lesion contrast associated with immature myelination. When a patient developed hydrocephalus or underwent shunt placement or epilepsy surgery we selected the last brain MRI before such interventions.

## MRI findings in TSC

Cortical tubers are not the only findings of TSC in brain MRI [5]. Subependymal nodules and white matter radial migration lines were also present in some of our MRI images. However, since tubers were present in all images labeled as TSC and not present in any of the images labeled as controls, we expected the CNN to detect tubers as the pattern that differentiated TSC patients from controls. On brain MRI, tubers are moderately well-circumscribed areas of increased signal intensity on T2-weighted and fluid-attenuated inversion recovery (FLAIR) images. The cortex overlying the tuber may have features of malformation, and the gray-white matter differentiation is reduced [5].

## MRI sequences, image labeling, and division into training, validation, and testing

We only selected two-dimensional axial T2 and FLAIR sequences for both patients with TSC and controls. MRIs were collected for clinical reasons. The objective of the study was to develop a deep learning algorithm to detect tubers in MRI images (not to identify from patients with TSC from any image, as not all individual slices from patients with TSC contain lesions). For the MRI of each patient with TSC, a pediatric neurologist (ISF) selected several MRI slices in axial planes with obvious tubers in them. For the controls, the same pediatric neurologist selected MRI slices at approximately the same level in the brain that for patients with TSC. The images of 138 patients (69 TSC and 69 controls) were used for training the model, of 40 patients (20 TSC and 20 controls) were used to validate the model, and of 50 patients (25 TSC and 25 controls) were used to test the model. There was no patient overlap between the training, validation, and test sets. We developed and validated the model based only on labels derived from routine clinical radiology reports to demonstrate that these models can be developed with data generated from routine clinical care. The labels (TSC or control) for the images in the testing set were independently confirmed by a clinical neuroradiologist (EY).

## Minimizing overfitting

CNNs are complex mathematical functions with a huge number of parameters, which allow them to fit well complex datasets but, at the same time, makes them prone to fit the data too well with poor generalization. We used several techniques to minimize overfitting:1) keeping training, validation, and test sets completely independent of each other (held-out cross-validation, as is the standard approach in CNN); the test set consist of data the final deep learning model was never exposed to before and, therefore, it is a good evaluation of how well the final deep learning model will do on new data, 2) using random noise, batch normalization, dropout, and global average pooling in the CNN architectures, and 3) using data augmentation (https://ivansanchezfernandez.github.io/TSC_supplementary_methods/).

## Data augmentation

We augmented the training set by creating approximately 4 copies of each original image by randomly allowing shifts, horizontal flips, and rotations. Data augmentation of the training set is a standard training approach in CNN which makes CNNs more robust to the essential features of each class and less sensitive to particularities of the individual images used for CNN training such as laterality (left or right), location of the tuber in the image (for example, upper part of the image or lower part of the image), rotation, etc. [13]. In addition, data augmentation of the training set has a crucial role to obtain good CNN performance using relatively small datasets like the present one, effectively increasing the size of the training set for the CNN to learn without increasing the number of subjects [13]. We augmented our training data with Image data generator from Keras [14] using the following parameters: rotation range: 30 degrees, width and height shift range: 0.1, shear range: 0.2, zoom range: 0.25, we allowed horizontal flip; and we used nearest neighbors as the fill mode. The validation and test sets were not augmented.

## Variables

The primary outcome was the accuracy of the deep learning algorithm to correctly classify the MRI image as TSC or control in the test set (using the best performing CNN model in the validation set, as explained below). The secondary outcome was the ability of the deep learning algorithm to detect tubers within each image in the test set.

## Model development

CNNs process data in the form of arrays. In the case of two-dimensional images, each array is three-dimensional with two dimensions for width and height and one dimension for the three color channels (red, green, and blue), even when the MRI image is grayscale. Each numerical value in the three-dimensional array represents the pixel intensity in the three color channels to construct the grayscale MRI image. We downscaled the input size of the images to 224x224x3 and normalized pixel values between 0 and 1, which are standard transformations for input data used in CNNs. A CNN has an architecture with several layers so that the inputted image is transformed in several steps to eventually yield a class prediction [15]. The main types of layers are convolutional layers, pooling layers, and dense (fully-connected) layers. Convolutional layers "scan" different areas of the image trying to find spatial patterns, for example, edges. Pooling layers ensure that the local conjunction of features from the previous layers are detected regardless of their location in the image (translation and rotation invariance). The convolutional and pooling layers automatically perform most of the feature extraction and feature transformation and, towards the end of the CNN architecture, the image is

flattened to a one-dimensional vector. Afterward, fully-connected layers try to translate the feature vector into probabilities of the original image belonging to one of the classes [15]. Each CNN has thousands to millions of parameters to be tuned iteratively. These parameters have no direct interpretability, they are just weights in a complex mathematical function with no intuitive interpretation. During the forward pass the initial weights yield predictions. These predictions are compared with the ground-truth image labels and an error measure is calculated. The error is "backpropagated" with partial derivatives, so that the "responsibility" of each parameter in the error is calculated and the parameter is slightly changed in the direction that reduces that error. Through multiple forward passes and backpropagation steps, the parameters are tuned to optimize classification [15]. We tried three different CNN architectures: 1) Tuberous sclerosis complex convolutional neural network (TSCCNN), a relatively simple architecture that we developed with 4 blocks, each of them consisting of several convolutional layers followed by a pooling layer, and a final block of fully-connected layers (https://ivansanchezfernandez.github.io/TSC_supplementary_methods/), 2) InceptionV3, a popular architecture within the family of CNNs that parallelize computations in a split-transform-merge approach to increase depth and improve accuracy while keeping computations efficient [16, 17], and 3) ResNet50, a popular architecture within the family of residual CNNs that allow very deep CNNs by using blocks of layers that behave like relatively shallow classifiers and work together as an ensemble to produce a very good classifier [18–20]. The initial weights for TSCCNN were random weights, while the initial weights for InceptionV3 and ResNet50 were the transfer learning weights used during the ImageNet competition [21]. Transfer learning is the improvement of learning in a new task through the transfer of knowledge from a related task that has been already been learned and can potentially help performance in relatively small datasets [21]. Initialization weights are tuned in each step of backpropagation during training. However, initialization weights already trained to identify unrelated items in different images are easier to train than initial random weights because image recognition largely implies detection of edges and combination of these edges [21]. For all CNNs, we used Adam [22] as an optimizer with a learning rate of 0.00025, we used binary cross-entropy as loss function, and a batch size of 64 with 100 epochs.

## Model validation

The validation set evaluates how well the CNNs would generalize to images they were not trained on. We selected the most generalizable model as the one with the lowest binary cross-entropy loss in the validation set. Binary cross-entropy loss is a more granular and more stable measure of generalizability than accuracy, since it is a continuous measure as opposed to a dichotomous measure for classification.

## Model testing

While the purpose of model validation is to select the potentially most generalizable CNN, the purpose of model testing is actually testing generalizability in a completely new set of images: the test set. The primary outcome of classification accuracy was calculated as the proportion of images in the test set labeled correctly by the model. There was no patient overlap between the training, validation, and test sets, therefore the images in the test set came from patients who were not used in the training set or validation set and to which the CNNs were never exposed before.

## Model development versus clinical practice: Clinical cases

The development of a CNN entails obtaining observations (in our study, MRI images) and their labels (in our study, images containing tubers or images derived from a normal MRI) by a human to first train, then validate, and finally test the model. However, once the model has been trained, validated, and tested, its use in clinical practice is to assist a clinician in the recognition of new observations as belonging to one of the labels (diagnoses). In our study, that would mean recognizing whether an MRI image has tubers in it or not. As an illustration on how we envision the clinical use of our deep learning algorithm, we further evaluated the selected deep learning architecture in a series of 259 consecutive MRI T2 and FLAIR axial images from 6 additional patients with TSC. These images were not used in training, validation, or testing because of challenging characteristics, such as extremely subtle tubers or limited myelination.

## Model visualization

Machine learning can sometimes operate as a "black box" where it is not possible to know which features are used in the classification process. However, model visualization in CNN helps clarify the most relevant features used for classification. For example, a deep learning model used to diagnose disease from chest X-rays classified images as having pneumothorax because of the presence of a chest tube in the image [23]. Similarly, some deep learning models focused on areas in the image that indicated the origin of the image as hospital-based (portable X-ray) versus images obtained in the doctor's office to classify chest X-rays as normal or pathological [24]. Intuitive and graphical indications on how the CNN is making decisions helps users gain confidence in the model. Thus, to identify potential correct classifications based on incorrect features and to gain some intuition into the classification process, we identified image pixels most relevant for classification with gradient-weighted class activation maps (Grad-CAM) and saliency maps. Grad-CAM uses the gradient of the output category (TSC or control) to the last convolutional layer (the last layer with spatial information) to provide a coarse localization of the areas of most interest for classification (where the CNN is "preferentially looking at for making decisions") [25]. The objective of Grad-CAM and saliency maps is to make sure the CNN is classifying MRI slices based on what it is supposed to. The objective of this project was not to segment the patterns of interest within each image [26]. Saliency maps compute the gradient of the output category (TSC or control) to the original image, that is, they identify the pixels in the original image that, if changed, would modify most the probability of the image belonging to the class [27]. Roughly, class activation maps and saliency maps can be interpreted as "where the CNN is looking at" or, more precisely, which areas in the image are most important for classification into TSC versus control. Map visualizations are heatmaps of the gradients with the "hotter" colors representing the regions of most importance for classification.

## Statistical analysis and software

We summarized demographic and basic clinical data with descriptive statistics and calculated basic comparisons with rank sum Wilcoxon test for continuous variables and Fisher's exact test for categorical variables. For all CNNs, we used Adam [22] as an optimizer with a learning rate of 0.00025, we used binary cross-entropy as loss function, and a batch size of 64 with 100 epochs. We used Python version 3.6 [28] as programming language and within this language, its modules: Pandas [29], NumPy [30] and SciPy.Stats [31], to perform descriptive statistics, Keras [14] and scikit-learn [32] to build and analyze the CNNs, TensorFlow [33] as backend for CNNs, and Keras-vis for visualization of class activation maps and saliency maps [34]. We

used Jupyter notebooks to run and save the code and results [35]. We trained and validated our models with anonymized MRI images in a cloud computing system and tested the selected model in our local computer. We used a default threshold of 0.5: if the estimated probability was greater than 0.5, the MRI slice was classified as having tubers, otherwise it was classified as not having tubers.

# Results

## Demographic and clinical features

The TSC patients and controls included in the study were similar, except TSC patients were slightly younger than controls at the time of imaging (median 9.5 years versus 12.4 years) with no statistically significant difference between the groups (Table 1). The data were divided into 566 images for TSC training (69 patients), 130 images for TSC validation (20 patients), 210 images for TSC testing (25 patients), 561 images for control training (69 patients), 118 images for control validation (20 patients), and 226 images for control testing (25 patients). The training file, merging the TSC and control training images and after data augmentation contained 5,634 images. The validation file, merging the TSC and control validation images contained 248 images. The testing file, merging the TSC and control images contained 436 images.

## Validation set

The class activation maps in the validation set showed that all the models were classifying images based on the presence of tubers, rather than on other features. The best performing CNN in the validation set was InceptionV3 (loss: 0.5325) followed by ResNet50 (loss: 0.5400), and TSCCNN (loss: 1.0416). Therefore, we used InceptionV3 in the final model. Other performance values are summarized in S1 Table at https://ivansanchezfernandez.github.io/TSC_Supplementary_Table_S1 and also show that InceptionV3 performed better than the other CNN architectures in the validation set.

## Evaluation in the test set

InceptionV3 had an accuracy of 0.95 and an area under the receiver operating characteristic curve of 0.99 in the test set with a sensitivity of 0.95, specificity of 0.95, positive predictive value of 0.94, negative predictive value of 0.95, and F1 score (the harmonic mean of positive predictive value and sensitivity) of 0.95 (Table 2).

**Table 1. Demographic features in our population.**

|  | TSC | Control | Test statistic and p-value |
|---|---|---|---|
| **Age in years [median (p25-p75)]** | 9.5 (5–15.3) | 12.4 (6.9–15.7) | Wilcoxon rank sum test: -1.51 |
|  |  |  | *p*-value: 0.13 |
| **Sex (male:female)** | 64:50 | 61:53 | Fisher's exact test odds ratio: 1.11 |
|  |  |  | *p*-value: 0.79 |
| **Number of images per patient [median (p25-p75)]** | 8 (5–10) | 8 (7–8) | Wilcoxon rank sum test: 0.27 |
|  |  |  | *p*-value: 0.79 |

p25-p75: 25th and 75th percentiles.

**Table 2. Performance of InceptionV3 in the test set.**

| InceptionV3 | | Real classification | | |
|---|---|---|---|---|
| Accuracy: 0.95 AUC: 0.99 | | TSC | Control | |
| Predicted classification | TSC | 199 | 12 | PPV: 0.94 |
| | Control | 11 | 214 | NPV: 0.95 |
| | | Sen: 0.95 | Spec: 0.95 | F1: 0.95 |

**AUC:** Area under the receiver operator characteristic curve. **F1:** F1-score. **NPV:** Negative predictive value. **PPV:** Positive predictive value. **Sen:** Sensitivity. **Spec:** Specificity.

## Model visualization

Although the purpose of this study was not segmentation, which requires a different CNN architecture, the class activation maps and saliency maps showed that the deep learning algorithm was focusing on tubers to classify the images (Figs 1 and 2). All the figures with the

| Name | Affiliation |
|---|---|
| Mustafa Sahin, MD, PhD | Boston Children's Hospital, Harvard Medical School, Boston, MA |
| Jurriaan M. Peters, MD, PhD | Boston Children's Hospital, Harvard Medical School, Boston, MA |
| Simon K. Warfield, PhD | Computational Radiology Laboratory, Department of Radiology, Boston Children's Hospital & Harvard Medical School, Boston, MA |
| Monisha Goyal, MD | Department of Neurology, University of Alabama at Birmingham, Birmingham, AL |
| Deborah A. Pearson, PhD | Department of Psychiatry and Behavioral Sciences, McGovern Medical School, University of Texas Health Science Center at Houston, Houston, TX |
| Marian E. Williams, PhD | Keck School of Medicine of USC, University of Southern California, Los Angeles, California |
| Darcy Krueger, MD, PhD | Cincinnati Children's Hospital Medical Center, Cincinnati, OH |
| Ellen Hanson, PhD | Department of Developmental Medicine, Boston Children's Hospital, Boston, MA |
| Nicole Bing, PsyD | Department of Developmental and Behavioral Pediatrics, Cincinnati Children's Hospital Medical Center, Cincinnati, Ohio |
| Hope Northrup, MD | The University of Texas Health Science Center at Houston, TX |
| Bridget Kent, MA, CCC-SLP | Department of Developmental and Behavioral Pediatrics, Cincinnati Children's Hospital Medical Center, Cincinnati, Ohio |
| Sarah O'Kelley, PhD | University of Alabama at Birmingham, Birmingham, AL |
| Martina E. Bebin, MD, MPA | University of Alabama at Birmingham, AL |
| Rajna Filip-Dhima, MS | F.M. Kirby Neurobiology Center, Boston Children's Hospital, Harvard Medical School, Boston, MA |
| Kira Dies, ScM, CGC | F.M. Kirby Neurobiology Center, Boston Children's Hospital, Harvard Medical School, Boston, MA |
| Joyce Y. Wu, MD | Mattel Children's Hospital, David Geffen School of Medicine at University of California Los Angeles, CA |
| Stephanie Bruns | Cincinnati Children's Hospital Medical Center, Cincinnati, OH |
| Benoit Scherrer, PhD | Computational Radiology Laboratory, Department of Radiology, Boston Children's Hospital & Harvard Medical School, Boston, MA |
| Gary Cutter, PhD | University of Alabama at Birmingham, Data Coordinating Center, Birmingham, AL |
| Donna S. Murray, PhD | Autism Speaks |
| Steven L. Roberds, PhD | Tuberous Sclerosis Alliance |

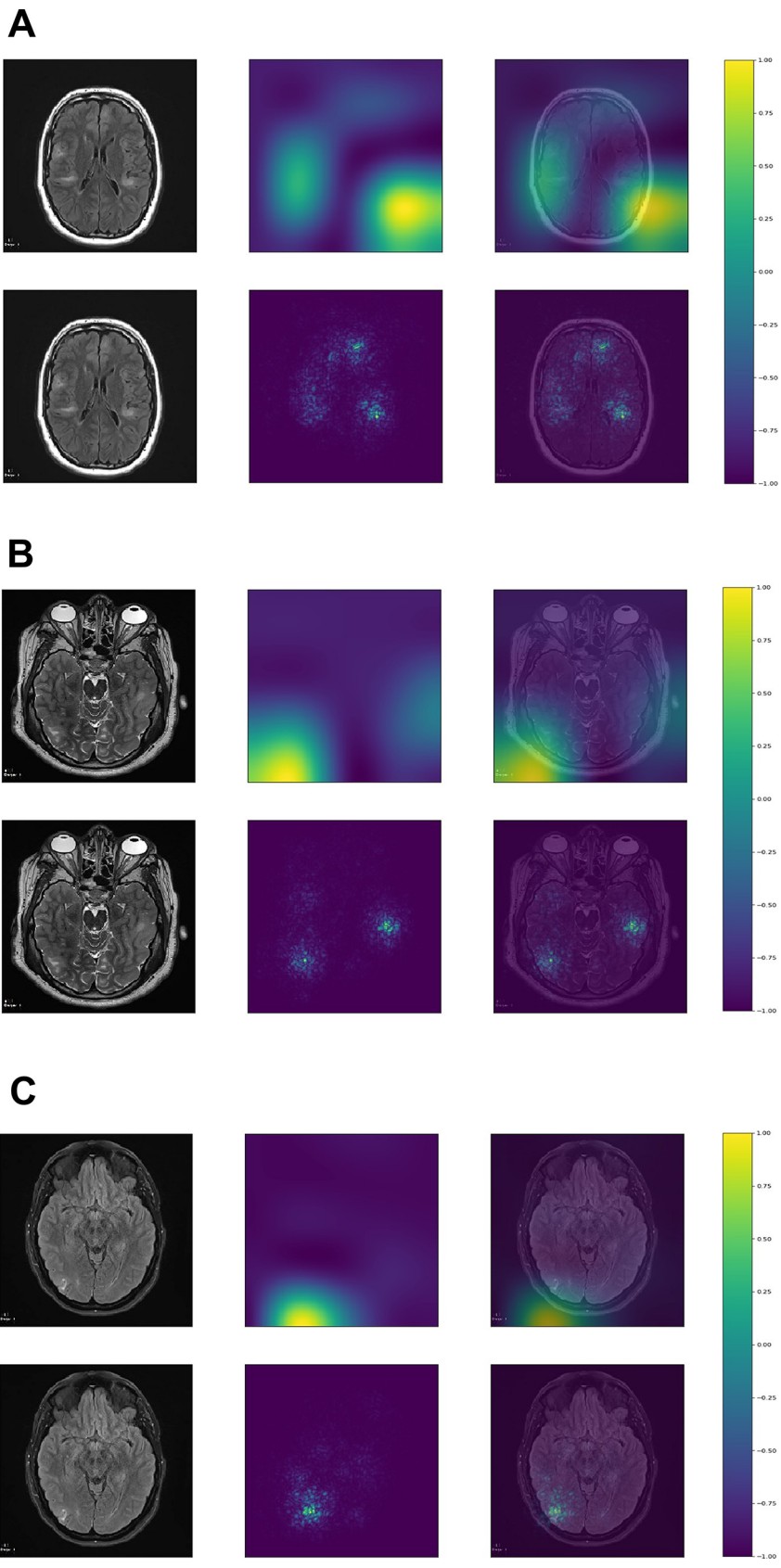

**Fig 1. Correctly classified images. A.** InceptionV3 was able to localize all or most tubers in this image with scattered and sometimes subtle tubers. **B.** InceptionV3 was able to localize the three relatively well-defined tubers in this image. **C.** InceptionV3 was able to localize the relatively well-defined tuber in this image. Although the image was classified as having tuber(s), the estimated probability was 0.71, as opposed to >0.99 for A and B. The first column represents the original image, the second column, the map, and the third column the map superimposed on the original image. The first row represents the gradient-weighted class activation map, and the second row represents the saliency map. Both gradient-weighted class activation maps and saliency maps visualizations are based on gradients. The gradient is the partial derivative of the loss function for each pixel in the image of reference (the last convolutional layer for gradient-weighted class activation maps and the original image for saliency maps). Gradient-weighted class activation maps use the gradient of the output category to the last convolutional layer (the last layer with spatial information). Saliency maps use the gradient of the output category to the original image. Both maps methods identify the pixels (in the last convolutional layer for gradient-weighted class activation maps and in the original image for saliency maps) that, if changed, would modify most the probability of the image belonging to the specific class (TSC or control). The resulting visualization is a heat map with values normalized between -1 (purple) and 1 (yellow) with hotter colors representing areas of greater importance for classification (see color bar at https://ivansanchezfernandez.github.io/TSC_heatmap_colorbar/). If you are not familiar with tubers, good examples can be found in Fig 1 in the Peters et al article summarizing neuroimaging in TSC [5]. A version of the images with arrows pointing to the tubers is available as S1 Fig at https://ivansanchezfernandez.github.io/TSC_Supplementary_Figures/.

Grad-CAM and saliency maps for the 436 testing images can be found at: https://ivansanchez fernandez.github.io/TSC_VisualizationI/#images, https://ivansanchezfernandez.github.io/ TSC_VisualizationII/#images, and https://ivansanchezfernandez.github.io/TSC_Visuali zationIII/#images. A graph with the colorbar can be found at https://ivansanchezfernandez. github.io/TSC_heatmap_colorbar/.

## Evaluation in clinical cases

To illustrate how readers can apply this deep learning model, we applied it to 6 new patients with TSC and particularly challenging radiological features such as very subtle tubers or age-related incomplete myelination. The deep learning model differentiated MRI slices with tubers from those without tubers with a sensitivity of 0.67, specificity of 0.68, positive predictive value of 0.84, negative predictive value of 0.46, and F1 score of 0.75. See the results and images at: https://ivansanchezfernandez.github.io/TSC_TestCases/.

## Interactive model

This deep learning model, the same as the one we used, is packaged in an App so it will produce the same results if exposed to the same images. This App avoids privacy challenges because it has no images on it, just the final model that users can test with their own images. We encourage readers to download and use the App on their own set of TSC test cases. Step-by-step use instructions available for Windows at: https://ivansanchezfernandez.github.io/ TSC_TuberFinder_Windows/ and Apple at: https://ivansanchezfernandez.github.io/TSC_ TuberFinder_Apple/. This system does not need to train the model, only test it with new images and, therefore, does not need a computer with graphics processing unit (GPU) and takes approximately one minute per image.

## Discussion

This study shows that deep learning algorithms for recognizing patterns in brain MRIs can perform very well, even when the number of patients and images are relatively small as is the case for rare neurological conditions. Further, this study shows that deep learning models trained in a cloud computing environment can be made portable to deploy on local computers avoiding many hurdles related to privacy. Our best performing deep learning algorithm (InceptionV3) detected MRI slices and areas of interest within each slice at a level similar to a

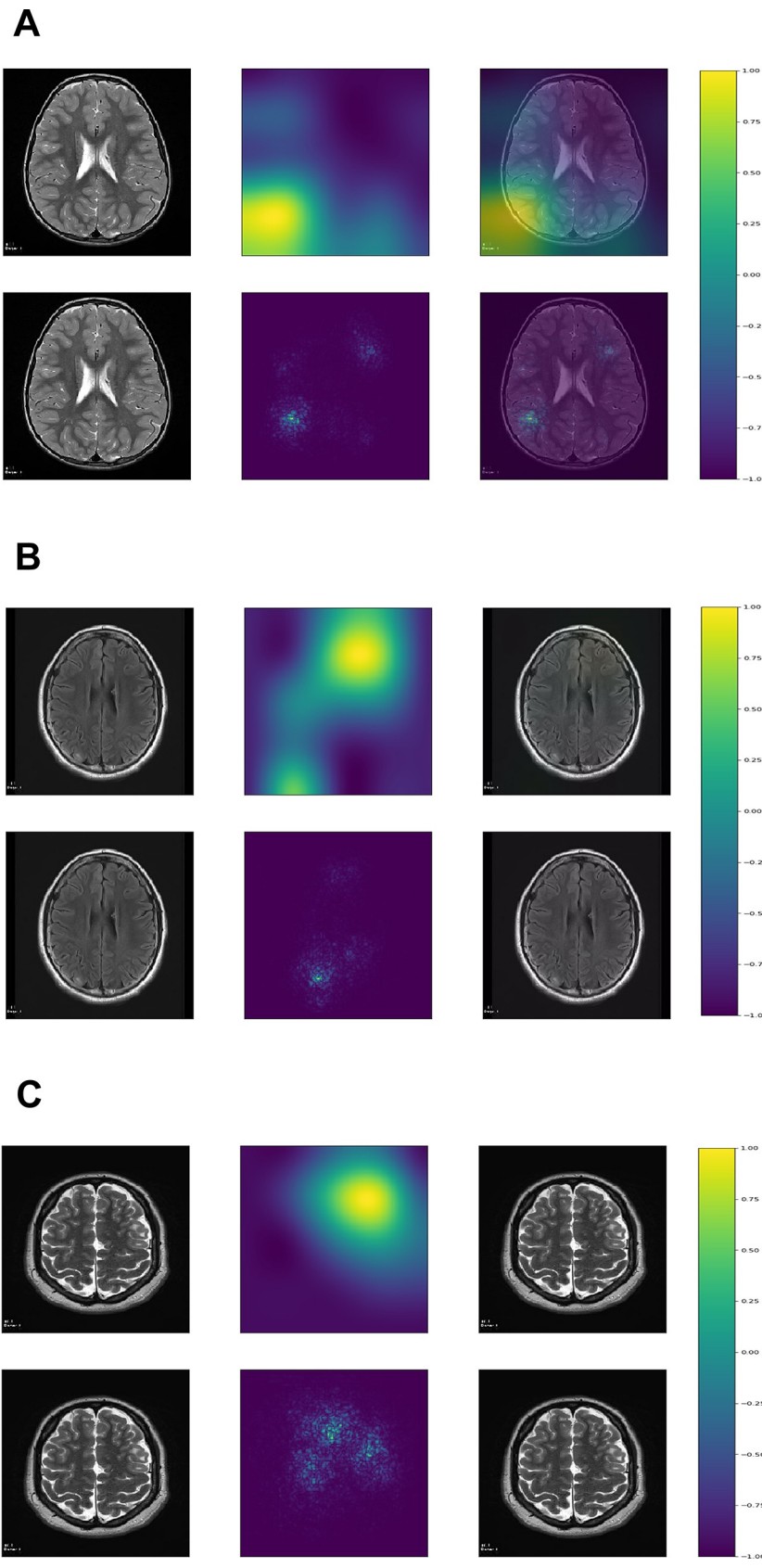

**Fig 2. Incorrectly classified images.** We would like to emphasize that incorrectly classified images represented only approximately 5% of the test set, but they sometimes provide insights into the reasons for misclassification. **A.** InceptionV3 classified this image as having tuber(s) with an estimated probability of 0.82, although it belonged to a control patient. The maps suggest a focus on prominent vascular spaces in the white matter suggestive of radial migration lines. **B.** InceptionV3 classified this image as having no tuber(s) despite the radiologist-confirmed subtle tuber in the right occipital region. The maps show a focus in the right region, but the model estimated a probability of having tuber(s) of only 4%. **C.** Although this occurred in a tiny minority of images, this image shows that sometimes the tuber is completely missed and the focus of the maps is not necessarily informative. The estimated probability of having tuber(s) was less than 1%. The first column represents the original image, the second column represents the map, and the third column represents the map superimposed on the original image. The first row represents the gradient-weighted class activation map, and the second row represents the saliency map. Both gradient-weighted class activation maps and saliency maps visualizations are based on gradients. The gradient is the partial derivative of the loss function for each pixel in the image of reference (the last convolutional layer for gradient-weighted class activation maps and the original image for saliency maps). Gradient-weighted class activation maps use the gradient of the output category to the last convolutional layer (the last layer with spatial information). Saliency maps use the gradient of the output category to the original image. Both maps methods identify the pixels (in the last convolutional layer for gradient-weighted class activation maps and in the original image for saliency maps) that, if changed, would modify most the probability of the image belonging to the specific class (TSC or control). The resulting visualization is a heat map with values normalized between -1 (purple) and 1 (yellow) with hotter colors representing areas of greater importance for classification (see color bar at https://ivansanchezfernandez.github.io/TSC_heatmap_colorbar/). If you are not familiar with tubers, good examples can be found in Fig 1 in the Peters et al article summarizing neuroimaging in TSC [5]. A version of the images with arrows pointing to the tubers (except for 2A which had no tubers) is available as as S2 Fig at https://ivansanchezfernandez.github.io/TSC_Supplementary_Figures/.

clinical radiologist. This system could provide diagnostic support in cases of suspected TSC, in non-academic settings where exposure to rare diseases is limited, and in low-resources areas with a limited number of expert medical specialists. At this time, however, clinical applicability is limited to discrimination of images with tubers from those without such lesions.

One of the major advantages of CNNs is that there is no need to specify the features that allow tuber recognition: CNNs automatically learn by examining multiple examples with the pattern and multiple examples without the pattern [15]. CNNs have revolutionized image classification and pattern detection within images in the last few years [15]. The ImageNet Large Scale Visual Recognition Challenge, popularly known as ImageNet, represents the benchmark for large-scale object recognition in computer vision and has allowed enormous progress in computer vision since 2012 [7]. In 2010 and 2011, the error rate of computer vision classification systems in ImageNet was above 25%, but the use of CNNs and enormous computing power with graphics processing units (GPUs) reduced that error below 10% in the period 2012–2014 [7]. Successive improvements in CNN architecture and optimization techniques reduced the error rate of CNNs at or below the error rate of human classification starting in 2015 [7].

Advances within the computer vision field to detect and localize common patterns in images such as dogs, cats, boats, cars, etc. were quickly translated to image recognition in medicine. Using more than 1 million retinal fundoscopy images, a team at Google developed and trained a CNN that was able to classify diabetic retinopathy at or above the level of ophthalmologists in two different testing sets [8]. Similarly, a team of researchers at Stanford University used 127,463 images to train and validate the InceptionV3 architecture (pre-trained with the ImageNet weights) and was able to detect and classify skin cancers with similar accuracy to dermatologists in a 1,942 biopsy-labeled test set [9]. In the field of Neurology, using a dataset of hundreds of thousands of head CT scans, a multicenter study from India was able to train a CNN to detect specific critical findings (intracranial hemorrhage and its types, fractures, midline shift, and mass effect) with an area under the receiving operator curve above 0.9 in most categories [11].

Unfortunately, these numbers of training images are not feasible in rare neurological conditions. For example, the largest multicenter TSC registry (TOSCA) contains clinical data for

2,093 patients, but no neuroimaging data [37], and the largest TSC multicenter study with neuroimaging data from the TSC Autism Centers of Excellence Research Network (TACERN) contains only 390 MRIs of 143 subjects [38]. Our set of images originated from a single hospital had 1,127 original images for training (566 in the TSC training set and 561 in the control training set), much smaller than in previous deep learning studies. However, data augmentation generated approximately 4 artificial new images per original image, and thus, the final training set of 5,634 images was enough to successfully train our CNNs as shown in the network convergence plots and achieve very good performance measures and localization of the tubers as shown by the Grad-CAM and saliency maps. We compared three commonly used CNN architectures: TSCCNN (a CNN architecture we developed based on the popular approach of several blocks consisting of convolutional layers followed by a max-pooling layer, and finished by a set of fully-connected layers and initialized with random weights), InceptionV3 (a CNN architecture developed by Google initialized with the ImageNet weights), and ResNet (a CNN developed by Microsoft initialized with the ImageNet weights which won the 2015 ImageNet challenge). Although these architectures are complex and the number of weights is very large, these CNN are relatively easy to train and apply to new image recognition tasks.

A major limitation of prior clinical applications of deep learning algorithms is that they are seldom applied outside the original study. The enormous computing resources for training a CNN requires a cloud computing environment, with resultant complications in maintaining privacy and confidentiality. Recently, a deep learning system to detect mammographic breast density was introduced into a clinical workflow, but still within the same hospital system that developed the CNN [39]. To overcome the portability limitation, we have developed an interactive standalone application with the best performing model (InceptionV3) to let readers test this method on their local images. Our model was developed in a cloud environment, but with anonymized images, not with identifiable full MRIs. An aspect in which our study is novel is that our application contains the trained model, but not the original images with which it was trained, overcoming privacy issues that result from sharing identifiable patient imaging data. This application will also allow clinicians and radiologists to easily apply a deep learning algorithm to their own patients' images inside their own hospital's computing environment without need to share patient data with a third party. We hope this standalone application will fuel interest in this approach and allow first-hand experience with how CNN technology may potentially impact future clinical practice. We believe that projects applying deep learning to medicine with a practical application bridge the world of clinical medicine and the world of computer science and may fuel interdisciplinary collaboration.

Neural networks are extremely complex mathematical functions that map raw inputs (images in this case) to outputs (image classification into TSC or control in this case) with minimal to no human guidance [40]. Their complexity also makes them one of the least interpretable techniques among machine learning algorithms [40]. Neural networks may make errors with the same or lower probability than human-driven medicine [8, 9, 11, 41], but their implementation remains slow because the rationale of incorrect classifications cannot be explained, raising complex liability issues [42]. Fortunately, important features from CNNs used for image classification can be depicted with Grad-CAM and saliency maps. These techniques can be loosely interpreted as "attention maps": where within the image the CNN is paying attention to classify this image into a certain category. Of note, these maps are not to be interpreted as lesion segmentation maps. Lesion segmentation requires training images with manually marked edges of the lesion and a different CNN architecture. In this particular study, Grad-CAM and saliency maps clearly show that CNNs are classifying images based on

the detection of tubers rather than on the detection of other spurious information that may coexist with tubers.

## Strengths and weaknesses

The present model achieved a classification performance close to that of the gold standard (a clinical neuroradiologist (EY) with formal training in radiology, neuroradiology, and pediatric neuroradiology and part of the epilepsy center at our institution) despite being subject to multiple constraints, which show its robustness: 1) the CNNs were trained on a relatively small dataset, 2) original images were two dimensional .jpg images and their resolution was reduced to 224x224 pixels, a much lower resolution than that used by radiologists to interpret MRIs and of lower image quality, 3) the CNNs were standard architectures developed for multipurpose large object recognition, that is, no specific architectural changes were made to try to fit the CNNs to the task of identifying patterns in MRI images, and 4) the TSC patients were heterogeneous in the number and appearance of tubers in MRI. Despite these challenges, our work shows that CNNs are able to achieve a classification performance close to the gold standard, which may make them able to guide attention to certain MRI slices and areas within each slice where tubers are more likely. CNNs offer the advantage of objectivity: the deep learning model outputs an estimated probability of an image having tubers, and consistency: the deep learning model will always return the same results when presented with the same image. Large repositories of MRI images analyzed with more refined CNNs may further increase sensitivity and specificity to a level where this approach can be used in routine clinical practice. It was not the objective of this study to perform image segmentation. In the future, if a set of MRI images with tuber borders delineated by a radiologist becomes available, other architectures may be used to actually segment tubers. CNN architectures specifically designed for automatic segmentation such as "U"-shaped architectures or fully convolutional neural networks specifically designed for segmentation, may further improve the localization of individual tubers within an MRI slice at the pixel level. Similar to some prior deep learning studies [8, 9, 43, 44], we performed a binary classification task to differentiate between MRI images with tubers and without tubers. The performance of this deep learning model when presented with confounding pathologies (tumors, white matter lesions in multiple sclerosis, etc.) is unknown. In the future, once we gather enough MRI images with other pathologies we aim to develop multiclass classifiers that recognize multiple different pathologies on brain MRI. The performance of this deep learning model (trained only on subjects 5 year old or older) was very good when presented with tubers in the context of a more immature myelination status, as shown by the very challenging test cases (https://ivansanchezfernandez.github.io/TSC_TestCases/).

Deep learning algorithms have been implemented in medicine mostly within the limits of research studies. Their application in routine clinical practice has been challenged by huge computational requirements and privacy concerns. By integrating a trained deep learning algorithm within a standalone application, we demonstrate that application into routine clinical practice and portability of trained CNNs is feasible. Further, as the deep learning algorithm within the application processes images in the local computer, it can be used with patient-protected information as long as the local computer is within a HIPAA-compliant healthcare environment.

## Conclusion

This study shows that CNNs trained on a relatively small dataset of manually selected low-resolution images is able to detect and localize tubers with a performance close to the gold

standard of the neuroradiologist in clinical cases. These results, obtained through data augmentation, serve as a model on how to prudently apply deep learning research algorithms, even when neurological conditions and images for training are rare. This study also shows a model of disseminating deep learning models trained locally to a global audience overcoming privacy hurdles.

## Acknowledgments

We are sincerely indebted to the generosity of the families and patients in TSC clinics across the United States who contributed their time and effort to this study. We would also like to thank the Tuberous Sclerosis Alliance for their continued support in TSC research.

The content is solely the responsibility of the authors and does not necessarily represent the official views of the National Institutes of Health.

We would like to acknowledge the TACERN co-investigators.

Mustafa Sahin, MD, PhD (mustafa.sahin@childrens.harvard.edu) and Darcy Krueger, MD, PhD (darcy.krueger@cchmc.org) are the lead authors authors at TACERN.

## Author Contributions

**Conceptualization:** Iván Sánchez Fernández, Edward Yang, Paola Calvachi, Marta Amengual-Gual, Joyce Y. Wu, Darcy Krueger, Hope Northrup, Martina E. Bebin, Mustafa Sahin, Kun-Hsing Yu, Jurriaan M. Peters.

**Data curation:** Iván Sánchez Fernández, Edward Yang, Paola Calvachi.

**Formal analysis:** Iván Sánchez Fernández, Paola Calvachi, Marta Amengual-Gual, Kun-Hsing Yu.

**Investigation:** Iván Sánchez Fernández, Edward Yang, Paola Calvachi, Kun-Hsing Yu.

**Methodology:** Iván Sánchez Fernández, Paola Calvachi, Marta Amengual-Gual.

**Project administration:** Iván Sánchez Fernández.

**Software:** Iván Sánchez Fernández, Kun-Hsing Yu.

**Supervision:** Iván Sánchez Fernández, Edward Yang, Paola Calvachi, Joyce Y. Wu, Darcy Krueger, Hope Northrup, Martina E. Bebin, Mustafa Sahin, Kun-Hsing Yu, Jurriaan M. Peters.

**Validation:** Iván Sánchez Fernández.

**Visualization:** Iván Sánchez Fernández, Edward Yang, Paola Calvachi, Kun-Hsing Yu.

**Writing – original draft:** Iván Sánchez Fernández, Paola Calvachi, Jurriaan M. Peters.

**Writing – review & editing:** Iván Sánchez Fernández, Edward Yang, Paola Calvachi, Marta Amengual-Gual, Joyce Y. Wu, Darcy Krueger, Hope Northrup, Martina E. Bebin, Mustafa Sahin, Kun-Hsing Yu, Jurriaan M. Peters.

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
