## [Decision Letter · Decision Letter 0]

11 Mar 2020

PONE-D-20-02069

Deep learning in rare disease. Detection of tubers in tuberous sclerosis complex

PLOS ONE

Dear Dr. Sanchez Fernandez,

Thank you for submitting your manuscript to PLOS ONE. After careful consideration, we feel that it has merit but does not fully meet PLOS ONE’s publication criteria as it currently stands. Therefore, we invite you to submit a revised version of the manuscript that addresses the points raised during the review process.

We would appreciate receiving your revised manuscript by Apr 25 2020 11:59PM. To enhance the reproducibility of your results, we recommend that if applicable you deposit your laboratory protocols in protocols.io, where a protocol can be assigned its own identifier (DOI) such that it can be cited independently in the future. For instructions see: http://journals.plos.org/plosone/s/submission-guidelines#loc-laboratory-protocols

We look forward to receiving your revised manuscript.

Kind regards,

Kaiming Li

Academic Editor

PLOS ONE

Journal Requirements:

2. Our internal editors have looked over your manuscript and determined that it is within the scope of our Digital Health Technology Call for Papers. This collection of papers is headed by a team of Guest Editors for PLOS ONE: Eun Kyoung Choe (University of Maryland, College Park), Chelsea Dobbins (University of Queensland), Sunghoon Ivan Lee (University of Massachusetts, Amherst), and Claudia Pagliari (University of Edinburgh).

The Collection will encompass a diverse range of research articles on digital health technologies ranging from technology design to patient care and health systems management.  Additional information can be found on our announcement page: https://collections.plos.org/s/digital-health-tech.

If you would like your manuscript to be considered for this collection, please let us know in your cover letter and we will ensure that your paper is treated as if you were responding to this call. If you would prefer to remove your manuscript from collection consideration, please specify this in the cover letter.

"Research reported in this publication was supported by the National Institute of Neurological Disorders And Stroke of the National Institutes of Health (NINDS) and Eunice Kennedy Shriver National Institute of Child Health & Human Development (NICHD) under Award Number U01NS082320. The content is solely the responsibility of the authors and does not necessarily represent the official views of the National Institutes of Health."

5. One of the noted authors is a group or consortium ACERN Study Group. In addition to naming the author group, please list the individual authors and affiliations within this group in the acknowledgments section of your manuscript. Please also indicate clearly a lead author for this group along with a contact email address.

6. Please upload a copy of Supplementary Table S1, D2; Figure S1; S2 which you refer to in your text on page 12, 14, 15.

Reviewers' comments:

Reviewer's Responses to Questions

**Comments to the Author**

1. Is the manuscript technically sound, and do the data support the conclusions?

Reviewer #1: Yes

Reviewer #2: Yes

2. Has the statistical analysis been performed appropriately and rigorously? 

Reviewer #1: Yes

Reviewer #2: Yes

3. Have the authors made all data underlying the findings in their manuscript fully available?

Reviewer #1: Yes

Reviewer #2: Yes

4. Is the manuscript presented in an intelligible fashion and written in standard English?

Reviewer #1: Yes

Reviewer #2: Yes

5. Review Comments to the Author

Reviewer #1: Artificial intelligence and deep learning have accelerated studies of automated diagnosis for human diseases. Some published studies have focused on static pictures (retinopathy, skin cancer, and echocardiography etc.), and a few researches involve three-dimensional CT images. This submitted study goes further than the above ones: it analyzes three-dimensional and more detailed MR images. The author choose a rare disease with limited data to verify the feasibility of a deep learning algorithm method. It is a challenging and innovative study indeed, but I have several suggestions for the authors.

1, In real clinical practice, the logical flow of a diagnostic procedure is: (1) screening abnormalities from normal, then (2) performing differential diagnosis (tumor, FCD, etc.) from abnormalities to make a final diagnosis. In this manuscript, however, the logical flow is: （1）a specialist defines a specific disease (TSC) and completely normal controls respectively, then (2) a deep learning algorithm attempts to distinguish the two groups. As the gap between the two different logical flows could possibly hinder clinical application, I suggest the author to clarify the rationality of his research method further.

2, In general, deep learning uses large scale data, and the data are raw and unpicked, like the head CT study (S Chilamkurthy, Lancet 2018). In this manuscript, the author himself picked different specific slices (5-10 slices) from each patients’ MR scans (Line 117, &256) to bring into the CNN procedures. Whether this manual selection is the reason why deep learning can be done with a small number of cases. The method of deep learning after manual screening, it may be a feasible solution to produce results in the short term, but it might delivery misleading information and have restricted clinically applicational prospect. The author should make further declaration.

3, The Conclusion (Line 54, &459) should be refined by adding restrictive attributive to deliver precise information. For example, it may be concluded that deep learning algorithms can distinguish previously manual screened TSC MR scans from normal MR scans. Or, deep learning can be prudently applied to a small but highly selected dataset in a rare neurological disorder.

To sum up, the author suggests a new idea for deep learning application, although there are still some places that need further clarification. I’d like to recommend the manuscript for publication after some revisions.

Reviewer #2: Introduction:

- The motivation behind this work was clearly stated that Tuberous sclerosis complex (TSC) has diverse characteristics between individuals. Automatic tuber detection through the brain MRI can improve diagnostic certainty due to limited number of medical specialists. Deep learning approaches have been approved with promising performance on image classification tasks in many medical applications with a huge number of training images. But it is still challenging in rare neurological disorder diagnosis.

- This paper aims to demonstrate Convolutional neural networks (CNNs) can be developed for detection of rare brain anomalies with a relatively small dataset, which points to the availability of deep learning with transfer learning on new medical imaging tasks with good performance and a good solution to overcome privacy problems.

- The problems related to the TSC diagnosis was clearly stated. But there is no data about the diagnosis performance by specialists. I think such kind of information and comparison might be useful to have a better understanding of the difficulty of this classification task by human. The significance of deep learning for medical applications could be highlighted.

Methods:

- Dataset part was well explained in detail, from the inclusion criteria for TSC patients, the criteria for the selection of images, MRI sequences and the train/validation/test datasets. It is important to re-implement the experiment.

- The data augmentation was introduced with detailed parameters, which is important to avoid overfitting. Three different CNN architectures were deployed. One model was trained from scratch and another two models were fine-tuned from the pre-trained model. All training configurations were provided for reference. The best models were obtained from the validation dataset and further verified on the test dataset. No patients data existed in both the validation and test datasets, with extra data for further verification. The model visualization part mainly focuses on the application of Grad-CAM and saliency map methods, which could interpret the classification procedure of CNN models vividly and intuitively.

- All data were divided into three part for training, validation and testing with same number of patients and normal controls. It is more convinced that if k-fold cross-validation could be used to verify the model performance. Especially for small datasets, the variance could be high on different subsets of data. And such kind of performance variance could also be interesting to evaluate the robustness of models.

Results and Discussion:

- The performance with the InceptionV3 was provided. Based on the model visualization, the salient regions were highlighted. And incorrectly classified samples also provided some insights of misclassification. Therefore, the feasibility of the deep learning models on MRI diagnosis tasks has been demonstrated. With the cloud-based training and local inference, the privacy problem could also be avoided.

- The InceptionV3 was selected as the final model due to the lowest validation loss. I think the ResNet50 had comparable performance, which should also be involved for the further experiment and comparison. It is more convinced if similar conclusions could also be drawn.

- The potential of image segmentation was also discussed. As an extension from the current classification task, the segmentation task is a promising solution for better understanding beyond the existing clinical knowledge and experience. But the availability of datasets, the huge computational requirements and privacy concerns should be always noted.

Conclusion:

- The feasibility and good performance of deep learning models on the TSC classification has been discussed. With limited number of data, the competitive performance has been achieved compared to the neuroradiologist. The cloud-based computation and local inference are promising for further applications.

6. PLOS authors have the option to publish the peer review history of their article (what does this mean?). If published, this will include your full peer review and any attached files.

Reviewer #1: No

Reviewer #2: No

---

## [Author Response · Author response to Decision Letter 0]

24 Mar 2020

Dear Dr. Li, dear reviewers,

Thank you for your kind consideration of our manuscript entitled “Deep learning in rare disease. Detection of tubers in tuberous sclerosis complex” (PONE-D-20-02069). We are grateful for the helpful comments and we have modified the manuscript in order to implement reviewers’ constructive suggestions. Below, please, find a detailed response to the comments.

All changes are shown as “tracked changes” mode and highlighted in a version of the manuscript uploaded as a supplementary file.

Thank you for reconsidering our manuscript for publication.

Sincerely, 

Iván Sánchez Fernández & Jurriaan Peters 

 

RESPONSE TO EDITORIAL COMMENTS

Thank you for the reminder. Some of the authors as part of the TACERN collaborative received funding to collect the data, although this analysis of the data was not specifically funded. To err on the side of caution, we identified these authors by their initials, and mention the funding support in the Funding Section on the title page. Iván Sánchez Fernández has recently received an Amazon Web Services Cloud Credits for Research in the amount of $9,700 in computational credits for his project on “Identification and localization of tubers in Tuberous Sclerosis Complex with deep learning convolutional neural networks”. We have added this new computational credit support to the cover letter and in the study funding section of the title page.

Response to this comment in the text: Cover letter. Study funding section in the title page.

 To enhance the reproducibility of your results, we recommend that if applicable you deposit your laboratory protocols in protocols.io, where a protocol can be assigned its own identifier (DOI) such that it can be cited independently in the future. For instructions see: http://journals.plos.org/plosone/s/submission-guidelines#loc-laboratory-protocols

We appreciate the opportunity to share our protocols, analyses, and code. We made our code and results publicly available in full at GitHub and Zenodo. However, we also deposited our code and results at protocols.io at dx.doi.org/10.17504/protocols.io.bdt3i6qn. 

Response to this comment in the text: links to github.com, zenodo.org, and protocols.io

We reviewed and made sure that the formatting met the PLOS ONE formatting requirements. 

Response to this comment in the text: All files.

2. Our internal editors have looked over your manuscript and determined that it is within the scope of our Digital Health Technology Call for Papers. This collection of papers is headed by a team of Guest Editors for PLOS ONE: Eun Kyoung Choe (University of Maryland, College Park), Chelsea Dobbins (University of Queensland), Sunghoon Ivan Lee (University of Massachusetts, Amherst), and Claudia Pagliari (University of Edinburgh).

The Collection will encompass a diverse range of research articles on digital health technologies ranging from technology design to patient care and health systems management. Additional information can be found on our announcement page: https://collections.plos.org/s/digital-health-tech.

If you would like your manuscript to be considered for this collection, please let us know in your cover letter and we will ensure that your paper is treated as if you were responding to this call. If you would prefer to remove your manuscript from collection consideration, please specify this in the cover letter.

We thank the editors for this consideration, but would prefer our manuscript to be considered as a regular article. We specified this in the cover letter. 

Response to this comment in the text: Cover letter.

"Research reported in this publication was supported by the National Institute of Neurological Disorders And Stroke of the National Institutes of Health (NINDS) and Eunice Kennedy Shriver National Institute of Child Health & Human Development (NICHD) under Award Number U01NS082320. The content is solely the responsibility of the authors and does not necessarily represent the official views of the National Institutes of Health."

We appreciate the opportunity to clarify this relevant aspect. Some of the authors as part of the TACERN collaborative received funding to collect this data, although this analysis of the data was not specifically funded. We identified these authors by their initials, and mentioned the funding support in the Funding Section on the title page. 

Response to this comment in the text: Title page.

Although we would like to share the magnetic resonance imaging (MRI) data publicly, these are clinical data from patients and there are legal and ethical restrictions on how to share them. The Institutional Review Board at Boston Children’s Hospital allowed us to use de-identified MRI data for this study because we are medical researchers who had specific training on the ethical use of medical data and will use these data only for the purposes of this research. The possibility of re-identification of subjects when using clinical data in this manner by trained medical researchers is remote. However, making the data publicly available to any user in the internet is not allowed by Institutional Review Boards. Users with no ethical training in the management of clinical data may use the MRI images for non-research purposes or may try to re-identify patients. That possibility will violate the patients’ rights. Interested researchers may request permission to the Institutional Review Board at Boston Children’s Hospital to obtain these data and, after appropriate verification of training in ethical management of medical data and an appropriate contract on how to use these data, the authors will make the data available. 

Following the editorial comment, we have expanded the “Data availability” subsection to specify the legal and ethical restrictions of clinical data and the Institutional Review Board responsible for these data. 

Response to this comment in the text: “Patients and methods” section, “Data availability” subsection (lines 258-260 in the track changes manuscript).

5. One of the noted authors is a group or consortium TACERN Study Group. In addition to naming the author group, please list the individual authors and affiliations within this group in the acknowledgments section of your manuscript. Please also indicate clearly a lead author for this group along with a contact email address.

Thank you for the opportunity to further detail the participants in the TACERN consortium. We have provided a list of the TACERN co-investigators in the acknowledgments section of the manuscript and have identified Dr. Mustafa Sahin, MD, PhD and Dr. Darcy Krueger, MD, PhD as the lead authors for this group. 

Response to this comment in the text: “Acknowledgements” section (lines 602-603 in the track changes manuscript).

6. Please upload a copy of Supplementary Table S1, D2; Figure S1; S2 which you refer to in your text on page 12, 14, 15.

We have now uploaded a copy of Supplementary Table S1, D2, Figure S1, and S2 to the protocols.io files. These supplementary data can also be found on the GitHub and Zenodo repositories.

 

RESPONSE TO REVIEWER’S QUESTIONS 

Comments to the Author

1. Is the manuscript technically sound, and do the data support the conclusions?

Reviewer #1: Yes

Reviewer #2: Yes

2. Has the statistical analysis been performed appropriately and rigorously? 

Reviewer #1: Yes

Reviewer #2: Yes

3. Have the authors made all data underlying the findings in their manuscript fully available?

Reviewer #1: Yes

Reviewer #2: Yes

4. Is the manuscript presented in an intelligible fashion and written in standard English?

Reviewer #1: Yes

Reviewer #2: Yes

RESPONSE TO REVIEWER 1

Artificial intelligence and deep learning have accelerated studies of automated diagnosis for human diseases. Some published studies have focused on static pictures (retinopathy, skin cancer, and echocardiography etc.), and a few researches involve three-dimensional CT images. This submitted study goes further than the above ones: it analyzes three-dimensional and more detailed MR images. The author choose a rare disease with limited data to verify the feasibility of a deep learning algorithm method. It is a challenging and innovative study indeed, but I have several suggestions for the authors.

1, In real clinical practice, the logical flow of a diagnostic procedure is: (1) screening abnormalities from normal, then (2) performing differential diagnosis (tumor, FCD, etc.) from abnormalities to make a final diagnosis. In this manuscript, however, the logical flow is: (1) a specialist defines a specific disease (TSC) and completely normal controls respectively, then (2) a deep learning algorithm attempts to distinguish the two groups. As the gap between the two different logical flows could possibly hinder clinical application, I suggest the author to clarify the rationality of his research method further.

We appreciate the opportunity to clarify this important point. Training a convolutional neural network (and other deep learning and machine learning models) requires labelled data. Labels classify observations (for example, MRI images) as belonging to some category (for example, an MRI with tubers or a normal MRI) and allow the model to “learn” during training the main features that make some observations (for example, MRI images) belong into some category. 

Once trained, the convolutional neural network is presented with new observations (for example, MRI images) and is tested on its ability to predict their label (for example, an MRI with tubers or a normal MRI) on these previously unseen observations. 

Therefore, extracting observations (MRI images in our case) and their labels (having tubers or belonging to normal MRIs) is a prerequisite to develop a neural network. However, once the model has been trained, validated, and tested, its clinical use follows the typical clinical flow of showing the model new data and expecting the model to provide a label (diagnosis) for it. 

Following the reviewer’s thoughtful comment, we have expanded on this relevant aspect in the newly created subsection “Model development versus clinical practice: clinical cases”.

Response to this comment in the text: “Patients and methods” section, “Model development versus clinical practice: clinical cases” subsection (lines 214-219 in the track changes manuscript).

2, In general, deep learning uses large scale data, and the data are raw and unpicked, like the head CT study (S Chilamkurthy, Lancet 2018). In this manuscript, the author himself picked different specific slices (5-10 slices) from each patients’ MR scans (Line 117, &256) to bring into the CNN procedures. Whether this manual selection is the reason why deep learning can be done with a small number of cases. The method of deep learning after manual screening, it may be a feasible solution to produce results in the short term, but it might delivery misleading information and have restricted clinically applicational prospect. The author should make further declaration.

We appreciate the reviewer’s feedback. The main reason for manually selecting slices is that in a situation where we would select normal MRI slices (without tubers) from a patient with tuberous sclerosis complex would misinform CNNs. Such images would prohibit training as the label for these slices would be “TSC+” because they are derived from patients with TSC, but there would not any features in the slices that allows them to be classified as belonging to the TSC group. In other words, the objective of the study was to develop a deep learning method to detect tubers in MRI slices, not to detect MRI slices (with or without tubers in them) from patients with tuberous sclerosis complex. 

Other reasons include privacy and confidentiality (3-dimensional brain MRIs are more easily re-identifiable than slices) and the structure of convolutional neural networks. Most convolutional neural networks have been developed to take as input two-dimensional images. Although there are convolutional neural networks to be used with 3 dimensional inputs, they are not as advanced as convolutional neural networks for 2 dimensional data. 

The final deep learning model can be applied to new MRI images as we showed with the test cases in our manuscript. This model is readily available for use on any new data in our standalone app that any user can try on their own 2D imaging data 

We recognize, however, the concern of the reviewer that the clinical applicability is limited to the user provide slices containing an abnormality. We have clarified this in the manuscript at two locations – in the “Patient and methods” section, and in the “Discussion” section, as outlined below. 

Response to this comment in the text: 

1 - “Patients and methods” section, “MRI sequences, image labeling, and division into training, validation, and testing” subsection (lines 127-129 in the track changes manuscript).

2 – “Discussion” section, lines 384-386

3, The Conclusion (Line 54, &459) should be refined by adding restrictive attributive to deliver precise information. For example, it may be concluded that deep learning algorithms can distinguish previously manual screened TSC MR scans from normal MR scans. Or, deep learning can be prudently applied to a small but highly selected dataset in a rare neurological disorder.

We agree that the conclusion should not overstate the clinical applicability, and that we should be precise about limitations of our trained CNN. Thus, following the reviewer’s thoughtful suggestion, we modified our conclusions to reflect that deep learning algorithms, once trained on manually screened MRI scans are able to differentiate normal from abnormal MRI slices and this can be prudently applied clinically to identify whether there are tubers in an MRI. (We would like to emphasize that the manual selection of images is performed only to provide representative examples during the training process, but once the algorithm is trained, the CNN model can recognize tubers in a set of unselected images as we showed in the clinical test examples). We agree, however, with the reviewer that the conclusion should better reflect that approach and modified the conclusions accordingly.

Response to this comment in the text: “Abstract” section (lines 63-65 in the track changes version of the manuscript), “Conclusion” subsection (lines 491-494 in the track changes version of the manuscript).

To sum up, the author suggests a new idea for deep learning application, although there are still some places that need further clarification. I’d like to recommend the manuscript for publication after some revisions.

We appreciate the helpful comments which helped us improve the quality of the manuscript.

RESPONSE TO REVIEWER 2 

- The motivation behind this work was clearly stated that Tuberous sclerosis complex (TSC) has diverse characteristics between individuals. Automatic tuber detection through the brain MRI can improve diagnostic certainty due to limited number of medical specialists. Deep learning approaches have been approved with promising performance on image classification tasks in many medical applications with a huge number of training images. But it is still challenging in rare neurological disorder diagnosis.

We are glad the reviewer appreciates and recognizes the challenge of deep learning in rare disorders. Deep learning approaches, although developed and used in many medical applications where a huge number of images are available, can be challenging to develop in rare neurological conditions where the number of available images is relatively small. Our approach is novel in that it shows that data augmentation can help train convolutional neural networks so that their performance is still very good even when the number of images available is not huge. We also appreciate the opportunity to emphasize that these models may be useful in areas of the world with a limited number of expert medical specialists. Following the reviewer’s thoughtful comment, we further emphasized these relevant aspects in the manuscript.

Response to this comment in the text: “Introduction” section, second paragraph (line 82 in the track changes version of the manuscript). “Discussion” section, first paragraph (line 384-386 in the track changes version of the manuscript).

- This paper aims to demonstrate Convolutional neural networks (CNNs) can be developed for detection of rare brain anomalies with a relatively small dataset, which points to the availability of deep learning with transfer learning on new medical imaging tasks with good performance and a good solution to overcome privacy problems.

We are grateful for the chance to further emphasize these relevant points. As the reviewer points out, our approach is novel because it shows that CNNs can be developed for detecting rare brain anomalies, even when the number of available images is relatively small. We emphasized deep learning with transfer learning and the ability to pack models in apps to overcome the privacy problems.

Response to this comment in the text: “Patients and methods” section, “Data augmentation” (lines 153-154 in the track changes version of the manuscript), and “Model development” subsections (lines 196-197 in the track changes version of the manuscript). “Discussion” section, fifth paragraph (lines 432-433 in the track changes version of the manuscript).

- The problems related to the TSC diagnosis was clearly stated. But there is no data about the diagnosis performance by specialists. I think such kind of information and comparison might be useful to have a better understanding of the difficulty of this classification task by human. The significance of deep learning for medical applications could be highlighted.

As the reviewer correctly points out, there is no formal data on the diagnostic performance by specialists. From our experience in clinical practice, detecting tubers and quantifying tuber burden is not difficult for trained neuroradiologists and pediatric neurologists. However, the benefit of developing CNN models to identify and quantify tuber burdens would be most helpful in non-academic settings and, especially, in low-resources areas where there is a limited number and limited access to expert medical specialists. 

Response to this comment in the text: “Discussion” section, first paragraph (lines 384 in the track changes version of the manuscript).

Methods:

- Dataset part was well explained in detail, from the inclusion criteria for TSC patients, the criteria for the selection of images, MRI sequences and the train/validation/test datasets. It is important to re-implement the experiment.

We agree with the reviewer that a rigorous approach to deep learning involves re-implementing the best model in a test set. We trained the model in the train set, validated its performance in the validation set, and the best performing model was tested in a test set. Because the training, validation, and test sets were independent (no patients in common between the three sets), the performance in the test set is a rigorous re-evaluation of the performance of the final model in data it was never exposed to previously. 

Response to this comment in the text: “Patients and methods” section, “Minimizing overfitting” subsection (line 142-144 in the track changes version of the manuscript).

- The data augmentation was introduced with detailed parameters, which is important to avoid overfitting. Three different CNN architectures were deployed. One model was trained from scratch and another two models were fine-tuned from the pre-trained model. All training configurations were provided for reference. The best models were obtained from the validation dataset and further verified on the test dataset. No patients data existed in both the validation and test datasets, with extra data for further verification. The model visualization part mainly focuses on the application of Grad-CAM and saliency map methods, which could interpret the classification procedure of CNN models vividly and intuitively.

The reviewer correctly summarized the main steps taken to minimize overfitting. We performed several steps to rigorously minimize overfitting. The visualization of the attention maps helps understand how the CNN model is making decisions. We further emphasized these aspects in the manuscript. 

Response to this comment in the text: “Patients and methods” section, “Model visualization” subsection (lines 230-231 in the track changes version of the manuscript).

- All data were divided into three part for training, validation and testing with same number of patients and normal controls. It is more convinced that if k-fold cross-validation could be used to verify the model performance. Especially for small datasets, the variance could be high on different subsets of data. And such kind of performance variance could also be interesting to evaluate the robustness of models.

Our approach of keeping independent training, validation, and test sets is also termed “held-out cross-validation”. Therefore, we have already performed cross-validation. 

We minimized overfitting and made our deep learning method robust by keeping sets of images (training, validation, and test) completely independent of each other, by making the convolutional neural network not prone to overfitting using addition of random noise, batch normalization, dropout, and global average pooling, and also by using data augmentation. These are the ways convolutional neural networks reduce overfitting and show robustness. 

Cross-validation is a machine learning technique typically used in other methods (such as random forests or support vector machines) with a relatively small number of parameters. Convolutional neural networks have thousands to millions of parameters and other methods are better to minimize overfitting and show robustness. Cross-validation is rarely used in convolutional neural networks and there are several reasons for that:

(1) Minimizing overfitting and showing robustness of convolutional neural networks is achieved with all the techniques shown above. For example, the dropout method or the ResNet50 architecture are conceptually similar to cross-validation in that the final model is an ensemble of different versions of the model trained with different data to maximize generalization.

(2) The way to evaluate the robustness of a deep learning model to new data is using a held out test set that is actually new data (the deep learning model never saw it). That is the approach used in convolutional neural networks applied to medical problems [Gulshan, V. et al. Development and Validation of a Deep Learning Algorithm for Detection of Diabetic Retinopathy in Retinal Fundus Photographs. Jama 316, 2402-2410, doi:10.1001/jama.2016.17216 (2016). Chilamkurthy, S. et al. Deep learning algorithms for detection of critical findings in head CT scans: a retrospective study. Lancet 392, 2388-2396, doi:10.1016/S0140-6736(18)31645-3 (2018).]. As long as the held out test set is a random sample of the original database, it is a perfectly valid way to evaluate performance in new data because the test set is actually new data. 

(3) In our particular database, cross validation in the training set. With 69 TSC patients and 69 control patients, a 10-fold cross-validation would be testing results on less than 7 TSC patients and less than 7 control patients in each fold, which is probably associated with a big random variability between the folds not necessarily reflecting model fit, but just small sample variability. Even when averaged this method does not appear to provide a clear advantage over the more standard split into training, validation, and test.

Response to this comment in the text: “Patients and methods” section, “Minimizing overfitting” subsection (lines 142-144 in the track changes version of the manuscript). 

For the interested reader, we would like to take this opportunity to clarify in more detail how this method is a held-out cross-validation, how overfitting is minimized and how the robustness of a convolutional neural network is evaluated in our paper. 

Convolutional neural networks are complex mathematical functions with a huge number of parameters, which allows them to fit well complex datasets but, at the same time, makes them prone to fit the data too well: overfitting. Overfitting occurs when the model recognizes patterns in the data but these patterns exist in the specific dataset, but they are not generalizable. There are several ways to avoid overfitting, some common to all convolutional neural networks and some specific to situations where there is a relatively small number of training examples. 

(1) Among the methods to minimize overfitting in convolutional neural networks in general, the most commonly used and powerful one is to completely separate training, validation, and test sets. With this method we actually evaluate generalizability because the convolutional neural network ability to detect patterns is tested in data it never saw before: the test set. Our convolutional neural networks were trained using only the images from the 69 TSC patients and 69 control patients. Each of the three convolutional neural networks learned only on the training set. The validation set with the images of 20 TSC patients and 20 control patients was used to track progress of the convolutional neural networks: see how well they were predicting on data they did not train on. That is, the validation set does not contribute to train any of the convolutional neural networks, but allows estimating how well they will generalize because these convolutional neural networks are tested on these validation data (data it never trained on). Further, the validation set serves another purpose when there are several convolutional neural networks being compared: once the convolutional neural networks have been completely trained and their parameters are final, the convolutional neural network with the best performance on the validation set is the one selected as the final model because it is the one with the best expected generalizability (it is performing the best with data it never trained on). Once the best (of the three) convolutional neural networks is selected, then it is tested on data it never saw before: the test set with 25 TSC patients and 25 control patients. Therefore, the performance of the selected convolutional neural network on the test set is a good estimation on how well it will generalize to new datasets because the test set is actually a new dataset it never saw before. There are no shared patients or images between the training, validation, and test datasets. The training, validation, and test datasets are completely separate from each other and that is one of the methods by which overfitting is avoided: we demonstrate generalizability of the final model by testing it in data it never saw: the test set. 

Explanations of the rationale for training, validation, and test sets can be found here https://deeplizard.com/learn/video/Zi-0rlM4RDs and here https://towardsdatascience.com/train-validation-and-test-sets-72cb40cba9e7. 

(2) Other general methods to minimize overfitting and make models robust in convolutional neural networks consist of avoiding the convolutional neural network to fit too well the training dataset. These include addition of random noise, batch normalization, dropout, and global average pooling. The global idea behind all these methods is making the training data “fuzzier” (note that this is a simplification of reality, the methods are more complicated) so that the convolutional neuronal is able to fit the features that are most characteristic of the pattern of interest, but not the noise that randomly appears in the training dataset and is not generalizable to other datasets. In particular, addition of random noise adds a layer to the initial architecture of the convolutional neural network so that the training input gets slightly modified and, therefore, it is more difficult for the convolutional neural network to memorize the training examples and makes it more robust and generalizable (see the Gaussian noise layers “model.add(GaussianNoise(0.2))” in our code https://ivansanchezfernandez.github.io/TSC_training_validation/). Batch normalization works by normalizing (subtracting the mean and dividing by the standard deviation) the outputs of a hidden layer so that the next hidden layer has inputs without extreme values helping training and, through regularization, avoiding overfitting (see the batch normalization layers “model.add(BatchNormalization())” and “BatchNormalization” in our code https://ivansanchezfernandez.github.io/TSC_training_validation/). Dropout works by randomly dropping some of the hidden units in the convolutional neural network at each training step so that the final convolutional neural network is an ensemble of different convolutional neural networks which makes it more robust to noise and minimizes overfitting (see the dropout layers “model.add(Dropout(rate = 0.5))” in our code https://ivansanchezfernandez.github.io/TSC_training_validation/). Global average pooling minimizes overfitting by reducing the total number of parameters in the model by reducing dimensions in the final layers by taking the average across some dimensions (see the global average pooling layers “GlobalAveragePooling2D()(x)” in our code https://ivansanchezfernandez.github.io/TSC_training_validation/). Explanations of these methods can be found here https://machinelearningmastery.com/train-neural-networks-with-noise-to-reduce-overfitting/, https://towardsdatascience.com/batch-normalization-in-neural-networks-1ac91516821c, https://machinelearningmastery.com/dropout-for-regularizing-deep-neural-networks/, and here https://alexisbcook.github.io/2017/global-average-pooling-layers-for-object-localization/.

(3) Finally, a method that is typically used when the number of training examples is relatively small is data augmentation. Data augmentation consists of synthetically creating artificial examples from the original training examples by randomly rotating, shifting, and zooming the original images. This method provide more examples for learning but it also has a regularization effect because the convolutional neural network cannot overfit noisy features (like location or size of the pattern of interest), but has to recognize the pattern of interest by its most characteristic features. 

In summary, we minimized overfitting through the above methods, and made our deep learning method robust by keeping sets of images (training, validation, and test) completely independent of each other, by making the convolutional neural network not prone to overfitting using addition of random noise, batch normalization, dropout, and global average pooling, and also by using data augmentation. 

Results and Discussion:

- The performance with the InceptionV3 was provided. Based on the model visualization, the salient regions were highlighted. And incorrectly classified samples also provided some insights of misclassification. Therefore, the feasibility of the deep learning models on MRI diagnosis tasks has been demonstrated. With the cloud-based training and local inference, the privacy problem could also be avoided.

Thank you for summarizing the main features of the “Results” section in our manuscript. We further emphasized that this App avoids privacy challenges because it has no images on it, just the final model.

Response to this comment in the text: “Results” section, “Interactive model” subsection (lines 367-368 in the track changes version of the manuscript). 

- The InceptionV3 was selected as the final model due to the lowest validation loss. I think the ResNet50 had comparable performance, which should also be involved for the further experiment and comparison. It is more convinced if similar conclusions could also be drawn.

Thank you for bringing up this important point. The usual approach in deep learning CNN is to extract only one model from the validation step and test it in the test set. In this way, improvements in performance from validation to testing cannot be attributed to “cherry picking” the model. However, the reviewer makes a very relevant point and following the reviewer’s thoughtful comment we tested the performance of ResNet50 on the test set with very good results as shown below. 

ResNet50

Accuracy: 0.94 AUC: 0.99 Real classification 

 TSC Control 

Predicted classification TSC 182 2 PPV: 0.99

 Control 28 287 NPV: 0.91

 Sen: 0.87 Spec: 0.99 F1: 0.92

Legend: AUC: Area under the receiver operator characteristic curve. F1: F1-score. NPV: Negative predictive value. PPV: Positive predictive value. Sen: Sensitivity. Spec: Specificity. 

The full code and results are present at https://ivansanchezfernandez.github.io/TSC_ResNet50/

Response to this comment in the text: As calculating the performance of a non-selected model in the test set is not standard in the deep learning CNN approach, we present these results only in the response to peer-review comments which will be publicly available for the interested reader through PLOS One open access standards. These results show the robustness of the methods.

- The potential of image segmentation was also discussed. As an extension from the current classification task, the segmentation task is a promising solution for better understanding beyond the existing clinical knowledge and experience. But the availability of datasets, the huge computational requirements and privacy concerns should be always noted.

We appreciate the opportunity to clarify this relevant aspect. We emphasized that image segmentation would require training images with manually marked edges of the lesion and a different convolutional neural network architecture. We aim to develop a study on lesion segmentation in the future and we are working on slowly building the appropriate dataset of manually marked images, but we do not presently have the data for that approach. 

Response to this comment in the text: “Results” section, “Model visualization” subsection (lines 302-303 in the track changes version of the manuscript). “Discussion” section, fifth paragraph (lines 432-433 in the track changes version of the manuscript). 

Conclusion:

- The feasibility and good performance of deep learning models on the TSC classification has been discussed. With limited number of data, the competitive performance has been achieved compared to the neuroradiologist. The cloud-based computation and local inference are promising for further applications.

We appreciate the helpful comments and we have emphasized all aspects discussed by the reviewer, which helped us improve the quality of the manuscript better through critical peer-review. 

6. PLOS authors have the option to publish the peer review history of their article (what does this mean?). If published, this will include your full peer review and any attached files.

Do you want your identity to be public for this peer review? For information about this choice, including consent withdrawal, please see our Privacy Policy.

Reviewer #1: No

Reviewer #2: No

Thank you, we used PACE and the images met PLOS specifications.

---

## [Decision Letter · Decision Letter 1]

14 Apr 2020

Deep learning in rare disease. Detection of tubers in tuberous sclerosis complex

PONE-D-20-02069R1

Dear Dr. Sanchez Fernandez,

We are pleased to inform you that your manuscript has been judged scientifically suitable for publication and will be formally accepted for publication once it complies with all outstanding technical requirements.

With kind regards,

Kaiming Li

Academic Editor

PLOS ONE

Additional Editor Comments (optional):

Reviewers' comments:

Reviewer's Responses to Questions

**Comments to the Author**

1. If the authors have adequately addressed your comments raised in a previous round of review and you feel that this manuscript is now acceptable for publication, you may indicate that here to bypass the “Comments to the Author” section, enter your conflict of interest statement in the “Confidential to Editor” section, and submit your "Accept" recommendation.

Reviewer #1: All comments have been addressed

Reviewer #2: All comments have been addressed

2. Is the manuscript technically sound, and do the data support the conclusions?

Reviewer #1: Yes

Reviewer #2: Yes

3. Has the statistical analysis been performed appropriately and rigorously? 

Reviewer #1: Yes

Reviewer #2: Yes

4. Have the authors made all data underlying the findings in their manuscript fully available?

Reviewer #1: Yes

Reviewer #2: Yes

5. Is the manuscript presented in an intelligible fashion and written in standard English?

Reviewer #1: Yes

Reviewer #2: Yes

6. Review Comments to the Author

Reviewer #1: The author clarified the necessity of labeled data and the difference between deep learning and clinical practice, by adding the “Model development versus clinical practice” paragraph. The author also elucidated the concern that clinical applicability is limited to the user provide slices containing an abnormality. The conclusion has been rewritten precisely to reflect the mechanism of the deep learning and the future application. Therefore I would like to recommend the manuscript for publication in Plos One.

Reviewer #2: (No Response)

7. PLOS authors have the option to publish the peer review history of their article (what does this mean?). If published, this will include your full peer review and any attached files.

Reviewer #1: No

Reviewer #2: No

---

## [Editor Report · Acceptance letter]

17 Apr 2020

PONE-D-20-02069R1 

Deep learning in rare disease. Detection of tubers in tuberous sclerosis complex 

Dear Dr. Sanchez Fernandez:

I am pleased to inform you that your manuscript has been deemed suitable for publication in PLOS ONE. Congratulations! Your manuscript is now with our production department. 

With kind regards,

on behalf of

Dr. Kaiming Li 

Academic Editor

PLOS ONE